# Thermal Characteristics of Slinky-Coil Ground Heat Exchanger with Discrete Double Inclined Ribs

**Teguh Hady Ariwibowo [1,2,*]** and **Akio Miyara [3,4]**

1    Graduate School of Science and Engineering, Saga University, 1 Honjo-machi, Saga 840-8502, Japan
2    Program Study of Power Plant System, DTME, Politeknik Elektronika Negeri Surabaya, Jl. Raya ITS, Surabaya 60111, Indonesia
3    Department of Mechanical Engineering, Saga University, 1 Honjo-machi, Saga 840-8502, Japan; miyara@me.saga-u.ac.jp
4    International Institute of Carbon-Neutral Energy Research, Kyushu University, Fukuoka 819-0395, Japan
*    Correspondence: teguhhady@pens.ac.id; Tel.: +81-707-648-6605

**Abstract:** The slinky ground heat exchanger (GHE) is the most widely utilized horizontal-type GHE, however, this GHE has a low curvature coil. The GHE has poor thermal mixing, especially at a low flowrate. At this flowrate, the coil heat exchanger has similar performance to a straight tube heat exchanger. Discrete double-inclined ribs (DDIR) are well known for their good thermal mixing by generating a vortex in straight tubes. In this paper, a numerical analysis of thermal performance for the plain coil and DDIR coil is discussed. It was found that the thermal performance of the DDIR coil was slightly higher than that of the plain coil in laminar flow. In turbulent flow, the DDIR coil was superior to the plain coil only in the first 149-min operation. The first 60-min analysis shows that in laminar flow, the average heat transfer rate in the plain coil is 59 W/m and in the DDIR coil is 60.1 W/m. In turbulent flow, the average heat transfer rate is 62 W/m, and the plain coil is 62.3 W/m. The copper DDIR coil material produced a better heat transfer rate than that of the composite and High-Density Polyethylene (HDPE). Sandy clay has the highest heat transfer rate. The influence of ground thermal conductivity on the performance of the GHE is more dominant than convection in the DDIR coil.

**Keywords:** discrete double-inclined ribs; low curvature coil; vortex generator; ground heat exchanger

## 1. Introduction

In recent decades, lack of energy, global warming, and air pollution are serious threats to the lives of living creatures in the world. The main trigger of global warming is carbon dioxide gas emissions mainly from fossil fuels [1]. Several attempts have been made to reduce the impact of $CO_2$ emissions by conserving energy [2–4]. To achieve this target, many researchers have focused on sustainable energy resources such as hydropower, geothermal energy [5,6], biomass, solar [7], and wind [8]. For the utilization of geothermal energy sources in shallow grounds, the ground source heat pump (GSHP) is a common alternative that is widely applied in various sectors in industries such as hot water production and heating or cooling air, both in commercial and domestic buildings.

The use of GSHPs results in efficient energy consumption, thus the installation of a GSHP has increased from 10% to 30% every year in 30 countries lately. The main advantage of the GSHP system is the high coefficient of performance compared to conventional heat pumps. The main reason is that the GSHP can use the ground as a heat sink in summer or heat source in winter [9,10]. Generally, there are two types of GSHP, including open-loop systems using groundwater or surface water directly and closed-loop systems with ground heat exchangers [11,12]. The closed-loop system is divided

into two types, namely the vertical type and the horizontal type, which is utilized depending on the geographical area and the availability of land.

An understanding of climate change and the thermal character of ground composition is essential in GHE installations. The design of the heat exchanger, which produces high thermal performance and is supported by the availability of land for installation, is taken into consideration. In addition, the GHE design has a balance between improved performance and financing costs.

Horizontal ground heat exchangers are preferred if the GHE is viewed from the installation. The GHE is only buried between 1 and 2 m from the ground surface. It does not require complicated equipment and skills in GHE installation [13]. The balance of the use of GSHPs in the summer and winter needs to be maintained. The ground thermal balance is vital for long-term usage of the GHE. The use of the GSHP system as a heater in winter is very profitable as a long-term investment [14–16].

Utilization of GSHPs in the summer and winter seasons can reduce the cost of energy more than methane heating systems or conventional air-conditioners. Moreover, the GSHP system produces lower pollutant emissions [17–19]. The use of the GSHP system has been widely applied in developed countries, but efforts to promote the GSHP system are critical. A good slinky GHE design can attract small companies and homeowners to implement the GSHP system.

The design and strategy to improve the performance of the slinky GHE is very necessary to achieve these goals. Mostly, the slinky coil has a curvature between 1.6 and 2.5 $m^{-1}$ which is considered as low curvature [20]. A low-curvature coil has similar characteristics to the straight tube in thermal- and hydrodynamics [21,22]. Modification of the pipe surface results in a more turbulent flow structure besides increasing the heat transfer area [23–28]. Corrugated plastic pipes were found to be able to increase heat transfer in Earth–air heat exchangers (EAHE). Meanwhile, corrugated [29] and twisted [30] tubes are used as heat transfer pipes for heat transfer at ground level.

The discrete double-inclined ribs (DDIR) tube is recommended to obtain heat transfer enhancement as a longitudinal vortex generator by several researchers [31–34]. They stated that the DDIR tube could carry out energy savings due to the heat transfer enhancement that is greater than the energy loss due to the pressure drop. We have conducted several studies on the effect of using DDIR on the thermal performance of the slinky coil under ideal and steady-state assumptions. Based on our findings, DDIR has the possibility of increasing the performance of the slinky coil [35–39]. This study aims to highlight the potential use of the DDIR coil in the slinky coil in ground heat exchangers under transient conditions in some different conditions.

## 2. Materials and Methods

### 2.1. Model Descriptions and Governing Equations

The computational domain is three-dimensional with a size of 6 × 1.4 × 5 m, as seen in Figure 1a. Due to computational limitations, the analysis is only one loop. The slinky GHE is buried 1.5 m from the ground surface. Then, the detailed geometry of the pipe can be seen in Figure 1b, where the slinky coil loop GHE consists of 2 straight tubes and a coil tube. The straight pipe is installed on the upstream and downstream of the coil. The length of the straight tube and coil tube is 0.7 and 3.1 m, respectively. The total axial length of one loop is 4.5 m. Details of ribs location on coil surface can be seen at Figure 2. The slinky coil diameter (D), coil pitch ($P_C$), and straight pipe are 1 m, 100 mm, and 700 mm, respectively. The angle of ribs ($\alpha$), axial ribs pitch ($P_R$), and ribs height (H) are 45°, 22.5 mm, and 1 mm, respectively. Coil dimensions and thermophysical properties of materials can be seen in Table 1.

This simulation uses ground at Saga University, Saga City, Japan. The topsoil layer is soft Ariake clay soil, which has a thickness of between 10 and 20 m with a maximum value of 30 m. Natural water content is 12 to 173% [40]. Soil samples were taken in the Saga City Fukudomi area, which consisted of clay from ground level to a depth of 15 m, sand, and sandy clay from 15 to 20 m and a water content of 30 to 150% which varied at different depths [41]. In this research, our concern in only the 5-m depth from the top ground surface; hence, we simulate heat transfer on clay. The thermophysical properties

of clay can be seen in Table 2. Sandy clay and sand are also tested to see the impact of ground thermal conductivity on GHE performance.

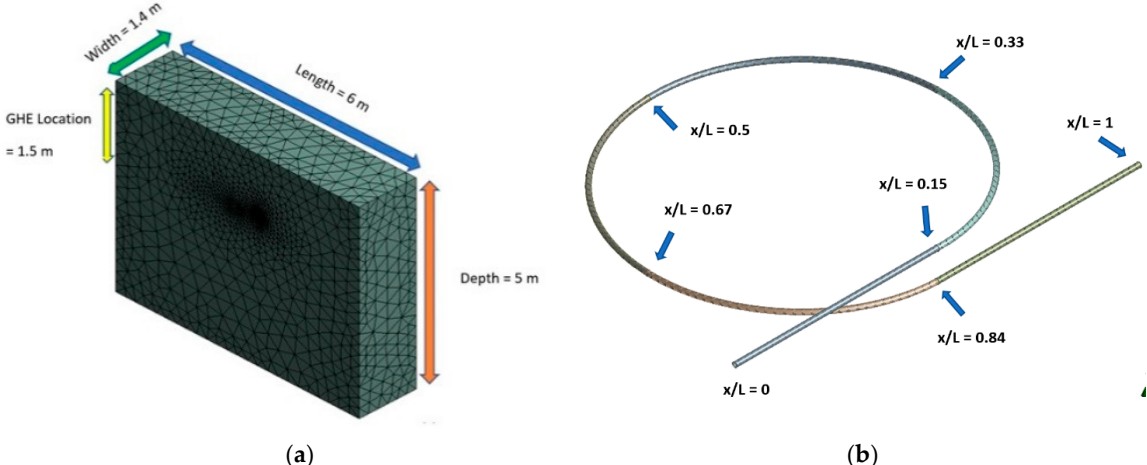

(**a**)    (**b**)

**Figure 1.** (**a**) Schematic diagram of the general computational domain; (**b**) general view of waterside computational domain and several cross-sections of the ground heat exchanger (GHE) for data collection.

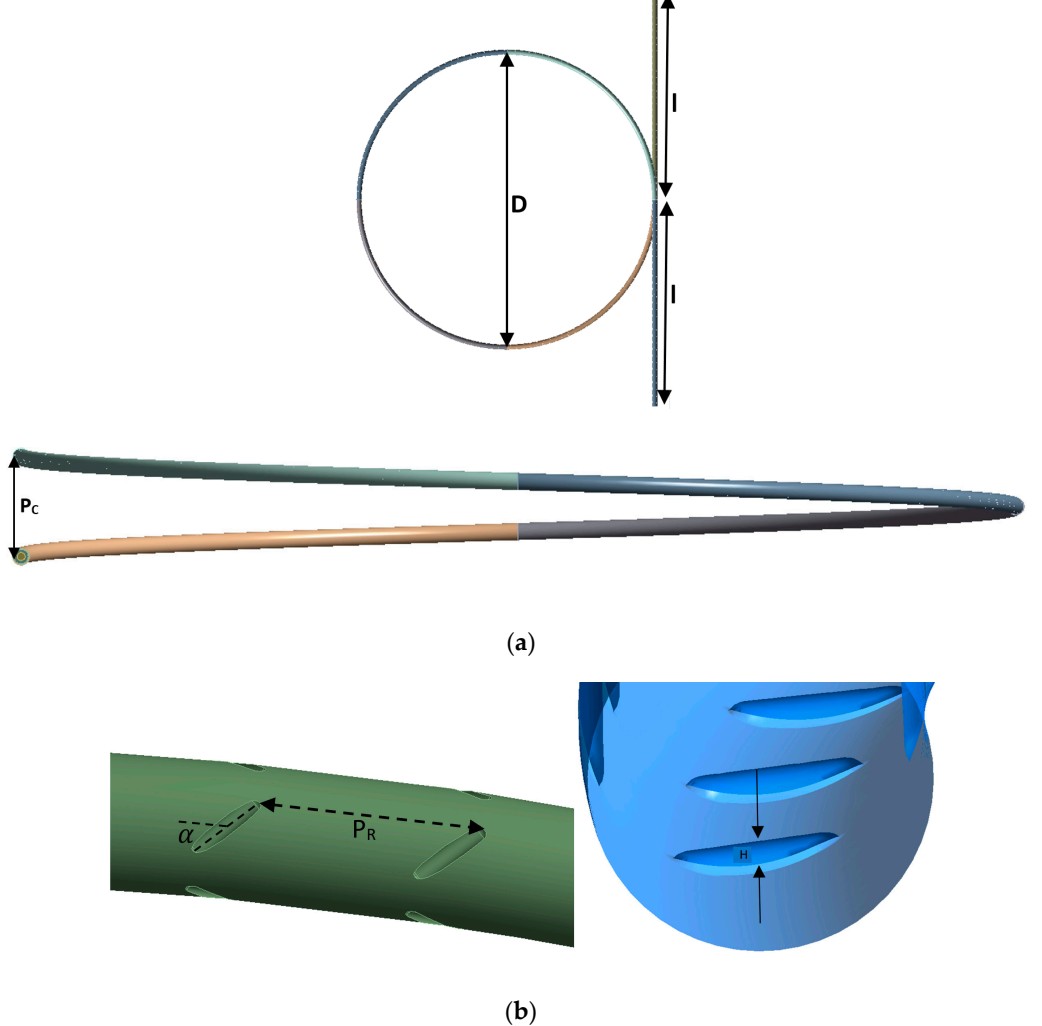

(**a**)

(**b**)

**Figure 2.** Configuration of the discrete double-inclined ribs (DDIR) coil. (**a**) Top view and side of coil, (**b**) location of ribs inside and outside the coil surface.

**Table 1.** Pipe sizing and thermophysical properties of materials.

| Pipe Material | Inner Diameter (mm) | Wall Thickness (mm) | Density (kg/m$^3$) | Specific Heat (J/kg·K) | Thermal Conductivity (W/m·K) |
|---|---|---|---|---|---|
| Composite: | | | | | |
| Copper (inner) | 14.9 | 0.65 | 8978 | 381 | 387.6 |
| LDPE [1] (outer) | - | 0.59 | 920 | 3400 | 0.34 |
| Copper | 14.9 | 1.24 | 8978 | 381 | 387.6 |
| HDPE | 14.9 | 1.24 | 955 | 2300 | 0.41 |

[1] LDPE: Low-Density Polyethylene.

**Table 2.** The properties of the ground.

| Parameters | Density (kg/m$^3$) | Heat Capacity (J/kg·K) | Thermal Conductivity (W/m·K) |
|---|---|---|---|
| Clay (water content: 27.7%) [1] | 1700 | 1800 | 1.2 |
| Sandy Clay (water content: 21.6%) [1] | 1960 | 1200 | 2.1 |
| Dry Sand (water content: 0%) [2] | 1815 | 620 | 0.3 |

[1] Taken from the JSME Handbook [42]. [2] Taken from Hailu et al. [43].

The main geometry and detailed meshing in the cross-section of the GHE can be seen in Figure 3. The computational domain is divided into three main parts, namely water and pipe, ground interface, and main ground. Meshing water and side pipes are less than 1 millimetre in size, while the main ground has a mesh size of 0.25 m, so a ground interface is needed, which is used as a link between the mesh in the water and pipe domain with the main ground. Figure 4a shows the general mesh around the coil. In Figure 4b, A and B are the fluid domains, C and D are the layers around the fluid domain, and E is the ground. The use of layers C and D can be seen in Table 3.

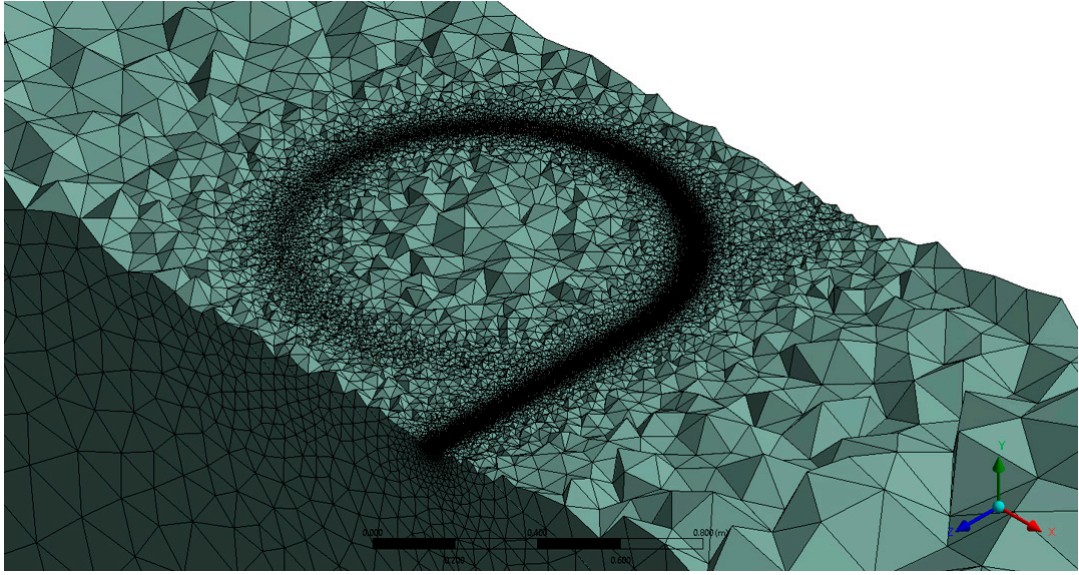

**Figure 3.** Mesh structure on the surrounding ground.

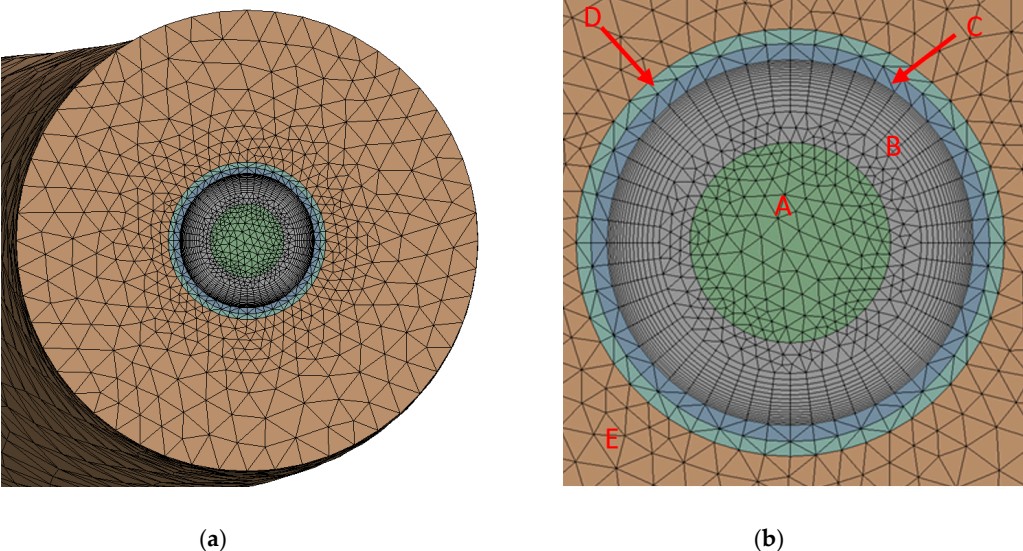

|  |  |
|:---:|:---:|
| (**a**) | (**b**) |

**Figure 4.** (**a**) Mesh structure on the coil and surrounding ground; (**b**) mesh structure water-side.

Flow simulation uses the Reynolds average Navier–Stokes (RANS) equation. This transient simulation solves the case of heat transfer and turbulent flow. In this case, the energy transport equation is used to solve the phenomenon of convection on the side of the water flow and conduction on the ground side around the GHE. The κ-ω SST turbulence model is utilized. To obtain more accurate results, the value of $y^+$ is 1. The time step used in the transient analysis was in minute basis. The details of the momentum equation, continuity, and energy transport in the fluid zone can be seen in the Ansys Manual [44]. Swirl strength ($\lambda_{ci}$) is utilized to find out the strength of the vortex. In this method, the strength of the vortex is calculated based on the velocity gradient tensor. Swirl strength uses the imaginary portion of the complex eigenvalue of the velocity gradient to interpret vortices [44,45].

*2.2. Boundary Conditions, Initial Conditions and Data Reduction*

A constant and uniform temperature of 29 °C is used at the top ground surface. In the bottom ground section, a constant heat flux of 65 mW/m² was used [46]. The temperature profile up to a depth of 5 m using experimental data was conducted on 1 July 2016, at Saga University, Japan [47]. For the cooling mode, the initial conditions of the temperature distribution on the ground are assumed to be the same as the temperature profile in the experiment. Temperature boundary conditions of far-field vertical grounds are set as constant temperature, while boundary conditions of ground that are close to the GHE are set as adiabatic. Velocity inlet and outflow are used as boundary conditions for inlet and outlet. Inlet water is assumed to be uniform velocity and a constant temperature at 27 °C. Determination of the critical Reynolds number for the flow in the coil using Ito's correlation [48] was as follows:

$$\text{Re}_{cr} = 20000\left(\frac{d}{D}\right)^{0.32} \tag{1}$$

According to the geometry size of the coil, the critical Reynolds number for the coil in this study is 5171. Based on $\text{Re}_{cr}$, the flowrate 2 L/min is a laminar flow with Re = 3406, and flowrate 4 L/min is a turbulent flow with Re = 6812. Analysis of the domain uses the temperature distribution profile using the equation illustrated in Figure 5.

$$T_y = 0.0148\, y^4 + 0.3366\, y^3 + 2.6865\, y^2 + 9.3082\, y + 29.62 \tag{2}$$

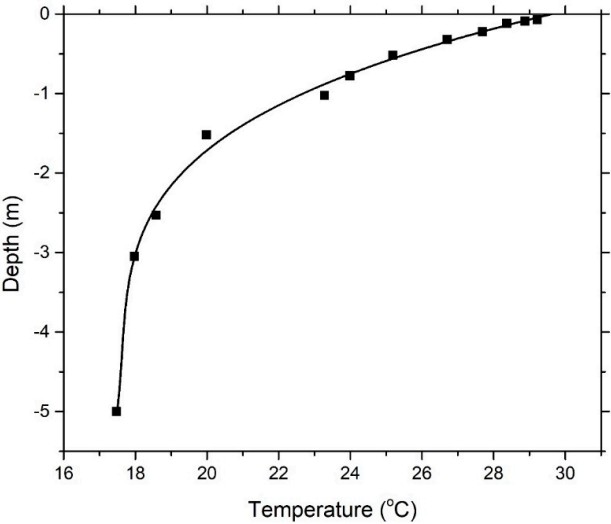

**Figure 5.** Ground temperature profile for the initial condition [46].

The heat transfer rate is calculated using the following equation:

$$Q = \dot{m}C_p \left(T_o - T_i\right) \tag{3}$$

where $\dot{m}$, $C_p$, $T_i$, and $T_o$ are the flow rate of the water (kg/s), the specific heat of the water (J/(kg·K)), and the inlet and outlet temperatures of the water, respectively. The heat transfer rate of the axial length of the coil or trench length can be calculated using the following equation:

$$\overline{Q} = \frac{Q}{L} \tag{4}$$

Local heat transfer coefficient is calculated by the following equation:

$$Q_l = \frac{Q''}{(T_b - T_w)} \tag{5}$$

where $Q''$, $T_b$, and $T_w$ are heat flux (W/m$^2$), bulk temperature (°C), and wall temperature (°C), respectively. Heat flux and wall temperatures were taken from the average value on the perimeter of the tube coil.

**Table 3.** Simulation conditions and fluid flow regime for all of the simulation models.

| Case | Flow Rate (L/min) | Pipe Material | Type Tube | Flow Regime | Ground |
|---|---|---|---|---|---|
| Case 1 | 2 | Composite | Coil Plain | Laminar [1] | Clay |
| Case 2 | 2 | Composite | Coil DDIR | Laminar [1] | Clay |
| Case 3 | 4 | Composite | Coil Plain | Turbulent [1] | Clay |
| Case 4 | 4 | Composite | Coil DDIR | Turbulent [1] | Clay |
| Case 5 | 2 | Copper | Coil Plain | Laminar [1] | Clay |
| Case 6 | 2 | Copper | Coil DDIR | Laminar [1] | Clay |
| Case 7 | 2 | HDPE | Coil Plain | Laminar [1] | Clay |
| Case 8 | 2 | HDPE | Coil DDIR | Laminar [1] | Clay |
| Case 9 | 2 | Composite | Straight Plain | Turbulent [2] | Clay |
| Case 10 | 4 | Composite | Straight Plain | Turbulent [2] | Clay |
| Case 11 | 2 | Composite | Coil Plain | Laminar [1] | Sand |
| Case 12 | 2 | Composite | Coil DDIR | Laminar [1] | Sand |
| Case 13 | 2 | Composite | Coil Plain | Laminar [1] | Sandy Clay |
| Case 14 | 2 | Composite | Coil DDIR | Laminar [1] | Sandy Clay |

[1] Critical Reynolds number of the coil is 5171. [2] Critical Reynolds number of the straight tube is 2300.

To understand the improvement of the GHE modification to the GSHP system, we use the net coefficient of performance which is proposed by Jalaluddin and Miyara [49], $COP_{net}$, in cooling mode as follows:

$$COP_{net} = \frac{Q_C}{L_{comp} + L_{pump}} \tag{6}$$

where $Q_C$ is the cooling rate, and $L_{comp}$ and $L_{pump}$ are the power inputs to the compressor and pump, respectively. If the DDIR coil increases the cooling rate by $Q'_C$ and pumping power by $L'_{pump}$, the net cooling COP becomes

$$COP'_{net} = \frac{COP_{net} + Q'_C/(L_{comp} + L_{pump})}{1 + L'_{pump}/(L_{comp} + L_{pump})} \tag{7}$$

The compressor power ($L_{comp}$) is assumed as constant. By using $COP_{net} < COP'_{net}$, then the following equation is obtained:

$$COP_{net\_cool} < \frac{COP_{net\_cool} + Q'_C/(L_{comp} + L_{pump})}{1 + L'_{pump}/(L_{comp} + L_{pump})} \tag{8}$$

Improvement of the system can be achieved with following equation:

$$Q'_C > L'_{pump} \tag{9}$$

The pumping power is given as the product of the volumetric flowrate V (m³/s) and pressure drop, $\Delta p$ (Pa/m).

$$L_{pump} = V \Delta p, \; L'_{pump} = V \Delta p' \tag{10}$$

From Equations (9) and (10), the following equation is obtained:

$$\frac{Q'_C}{Q_C} - \frac{V \Delta p}{Q_C} \frac{\Delta p'}{\Delta p} > 0 \tag{11}$$

where $Q_C$, $Q'_C$, V, $\Delta p$, and $\Delta p\prime$ are the cooling rate (W/m), an increase in the cooling rate (W/m), the volumetric flow rate (m³/s), the pressure drop (Pa/m), and an increase in the pressure drop (Pa/m), respectively. The Equation (11) is called Coefficient of Performance (COP) improvement factor.

For validation, the relative error (RE) between the experiment and simulation is calculated based on

$$RE = \frac{100}{N} \sum_{i=1}^{N} \left| \frac{Q_{exp} - Q_{sim}}{Q_{exp}} \right| \tag{12}$$

where $Q_{exp}$ and $Q_{sim}$ are the heat transfer rates from the experiment and simulation, respectively, and N is the number of data.

For analyzing the heat transfer rate, we use the temperature difference between the bulk and wall temperature as follows:

$$\Delta T_l = T_{bulk} - T_{wall} \tag{13}$$

### 2.3. Mesh Elements Independence and Model Validation

To conduct independent grid tests, three grids such as 15,819,114 (coarse), 19,455,734 (medium), and 24,987,160 (fine) elements are used as consideration. Case 3 is used for the grid independence test. After operating for 1440 min, the average heat transfer rate per length of trench was 42.1 W/m, 42.3 W/m, and 42.5 W/m, respectively, for 15,819,114, 19,455,734, and 24,987,160 elements. The deviation between the heat transfer rate of the coarse to fine mesh and medium to fine mesh is 0.96% and 0.5%, respectively. Hence, 24,987,160 grid systems are used in this study.

Figure 6 shows the heat transfer rate from experimental and simulated data. The experimental data are taken from a previous experiment at Saga University [46]. Due to the lack of experimental data, in this validation, experimental data are taken starting from the 99th minute of the beginning operation. In experiments, the inlet temperature fluctuates, so the heat transfer rate of the experiment fluctuates slightly. Water inlet temperature fluctuation in the simulation is similar to experimental data. It can be seen in the Figure 6 that the simulation and experimental data trends are the same and have almost the same value. The relative error is calculated based on Equation (12). The relative error heat transfer rate of the CFD and experiment is 7%.

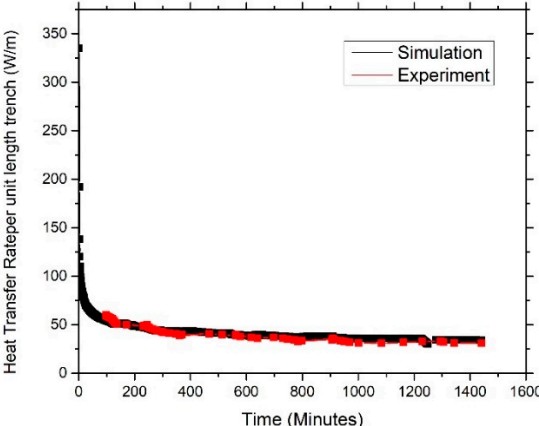

**Figure 6.** Comparison of heat transfer rate between experiment and simulation of the plain coil [47].

## 3. Results and Discussion

We discuss the performance of the DDIR coil and plain coil in several aspects. First of all, we present the streamlined pattern and vortex strength on both coils. The strength and location of the vortex on the coil are also described. Different flow structures could produce different frictional pressure drop characteristics. We only observe a pressure drop in cases 1, 2, 3, and 4 because the pipe wall is assumed as smooth. Second, the performance of the DDIR coil and plain coil heat transfer rate for 1440 min was studied. We investigated the operation of the first 60 min on some parameters such as temperature of the bulk, the temperature of the wall, the heat flux, and the heat transfer coefficient in several heat exchanger cross-sections. Energy loss due to a pressure drop and energy saving was also observed using COP improvement factors. Third, we observed the effect of the DDIR coil and plain coil performance on the thermal conditions of the soil at a specific location for 1440 min. In this section, we assume that the thermal property of the soil is uniform. Fourth, we describe the influence of the pipe material on the performance of the ground heat exchanger. Finally, we observe the dominance of the effect of soil conductivity on the thermal ground exchanger performance when compared to the use of the DDIR coil.

### 3.1. Flow Structure

The flow structure characteristics of the DDIR coil and plain coil need to be investigated before discussing the results. This flow analysis is quite useful to describe the flow in the coil. Figure 7 shows a 3D streamline on the plain and DDIR coils in cases 1 and 2. In the plain coil, the high velocity fluid particles occupy the location of the outer side of the coil. This phenomenon is caused by centrifugal force. This force also causes some water particles near the wall to move to the inner-side section. In the DDIR coil, high velocity fluid particles are almost uniformly distributed throughout the domain. The flow generated by ribs interferes with the centrifugal force of the coil. This incident causes the tendency of the flow of water particles to move distorted towards the outside coil before finally going to the inner-side coil. Thus, the particle path in the DDIR coil is further than that of the plain coil. The vortex strength analysis can be seen in Appendix A. Based on the flow characteristics, each pipe

has a different frictional pressure drop. The pressure drop in case 1 and case 2 are 219 and 395 Pa/m, respectively. Meanwhile, cases 3 and 4 are 676 and 1431 Pa/m, respectively. The straight plain, used as a pressure drop reference in calculating the COP improvement factor, has pressure drops for case 9 and case 10 of 65 and 205 Pa/m, respectively.

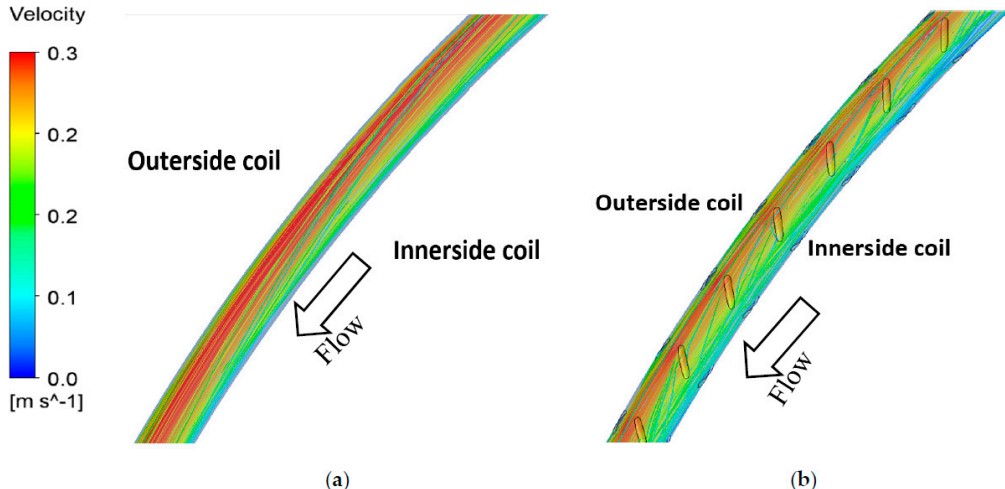

**Figure 7.** Top view of the streamline on the downstream coil for Re = 3406. (**a**) Case 1; (**b**) case 2.

## 3.2. Rib Effect on Heat Transfer Rate

The effect of the tube surface on the value of heat transfer in cases 1, 2, 3, and 4 as seen on Figure 8. Mostly, the heat transfer rate has peaked during the initial operation. Then, along with the increase in operating time, thermal performance starts to decline before it begins to decrease until the operation stops gradually. The GHE releases heat to the ground causing changes in the temperature of the surrounding ground during operation. The temperature difference between the ground and GHE decreases so that the amount of heat transfer decreases as the ground begins to heat up. This incident caused the heat transfer rate to drop dramatically due to the influence of dominant heat accumulation. Heat spreads to the surrounding ground; thus, the heat transfer rate continues to decrease at a low rate.

Based on the turbulent analysis, the DDIR coil has a higher performance than the plain coil in the first 149 min. In turbulent flow, for the first 149 min, the plain coil and DDIR coil heat transfer rates are 52.3 W/m. The DDIR coil heat transfer rate tends to be lower than the plain coil after the 149th minute until the last minute. On the other side, the heat transfer performance of the DDIR coil is larger than the plain coil from the beginning until the end of the time in laminar flow. Then, in laminar flow at the same operating time, the plain coil and DDIR coil heat transfer rates are 50 and 50.6 W/m, respectively.

In turbulent flow, the DDIR coil rejects heat in the ground higher than that of the plain coil at the beginning of the GHE operation. This phenomenon causes the heat of the ground around the GHE to experience a faster heating process than that of the plain coil. The ground is not able to provide enough thermal recovery, so the DDIR coil performance becomes low after 149 min. On the other hand, the DDIR coil tends to reject heat in the ground and is not too large at the beginning of the GHE operation in laminar flow. This phenomenon causes the heat accumulation in the ground to increase more slowly than that of turbulent flow. Therefore, the DDIR coil performance is still superior to the plain coil in the operation time.

Based on this phenomenon, it can be stated that the effect of the DDIR coil on turbulent flow has no significant increase in the heat transfer rate compared to the plain coil. In the flow structure analysis of the DDIR coil, the vortex strength is about four times larger than that of the plain coil. This flow structure should produce better thermal mixing. In Figure 8, DDIR coils do not show significant thermal performance. This phenomenon is caused by the limited ground thermal conductivity.

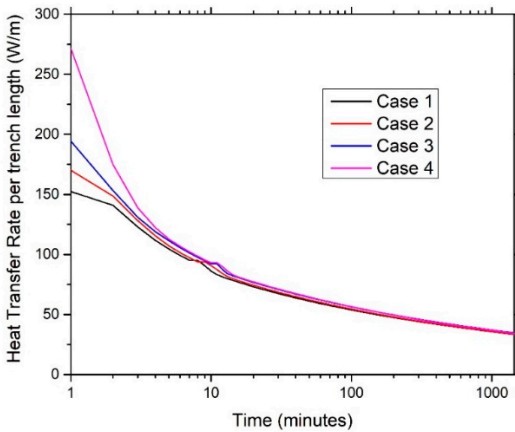

**Figure 8.** Comparison of the heat transfer rate performance of the plain coil and DDIR coil.

The average heat transfer rate was observed to evaluate the effect of the pipe modification on the GSHP system performance. Case 1 and case 2 have an average heat transfer rate of 40.8 and 41.1 W/m, respectively, whereas the average heat transfer rate for case 3 and case 4 is 42.5 and 42.4 W/m, respectively. The straight plain, used to reference the COP improvement factor calculation, has an average heat transfer rate in case 9 and case 10 of 13.8 and 14.7 W/m. COP improvement factors that are listed in Table 4 are always positive. In laminar flow, the DDIR coil is effective, whose COP improvement factor is 1.98. On the contrary, in turbulent flow, the plain coil is effective so that the COP improvement factor is 1.89.

**Table 4.** The criterion of COP improvement factor defined in Equation (11).

| GHE | $Q_{C-c}$ | $Q_{C-s} = Q_C$ | $Q'_C$ | V | $\Delta p_c$ | $\Delta p_s = \Delta p$ | $\Delta p'$ | Equation (11) |
|---|---|---|---|---|---|---|---|---|
| | W/m | W/m | W/m | m³/s | Pa/m | Pa/m | Pa/m | |
| Case 1 | 40.8 | 13.8 [a] | 27 | $3.333 \times 10^{-5}$ | 219 | 65 [a] | 154 | 1.96 |
| Case 2 | 41.1 | 13.8 [a] | 27.3 | $3.333 \times 10^{-5}$ | 395 | 65 [a] | 330 | 1.98 |
| Case 3 | 42.5 | 14.7 [b] | 27.8 | $6.666 \times 10^{-5}$ | 676 | 205 [b] | 471 | 1.89 |
| Case 4 | 42.4 | 14.7 [b] | 27.7 | $6.666 \times 10^{-5}$ | 1431 | 205 [b] | 1226 | 1.88 |

[a] Average heat transfer rate and pressure drop of straight tube at 2 L/min, [b] average heat transfer rate and pressure drop of straight tube at 4 L/min.

Figure 9 illustrates the distribution of the heat flux and heat transfer coefficient at axial locations x/L = 0, 0.15, 0.33, 0.5, 0.67, 0.84, and 1. Both parameters have share similarity with the trend graph. From x/L = 0 to 0.15, the heat flux decreases dramatically. In this part, we need to analyze by using wall temperature and bulk temperature distribution that is explained in Appendix A. We see that the heat flux trend has a relationship with $\Delta T_1$. $\Delta T_1$ at x/L = 0 is smaller than $\Delta T_1$ at 0.15. Then, in the part of x/L = 0.15 to 0.84, the heat flux does not show a significant difference. In this area, $\Delta T_1$ has almost the same value in all locations. Meanwhile, in the section from x/L = 0.84 to 1, the heat flux shows a significant decrease in the heat flux. $\Delta T_1$ at x/L = 0.84 is smaller than $\Delta T_1$ at x/L = 1. The heat flux values of the plain coil are higher than that of the DDIR coil at the same operation time because $\Delta T_1$ of the DDIR coil is smaller than $\Delta T_1$ of the plain coil. This means that the DDIR-coil produce better thermal mixing than that of plain coil. Then, heat transfer coefficients are calculated based on previous bulk temperature, wall temperature, and heat flux, which are described by using Equation (5). Generally, DDIR coils obtain heat transfer coefficients slightly greater than the plain coil along the axial length of the GHE both in laminar and turbulent flow. The heat transfer coefficient of the DDIR coil in turbulent flow is opposite to the heat transfer rate. This shows that actually the DDIR coil has better performance, but the dominance of the influence of soil conductivity is stronger than convection in the GHE.

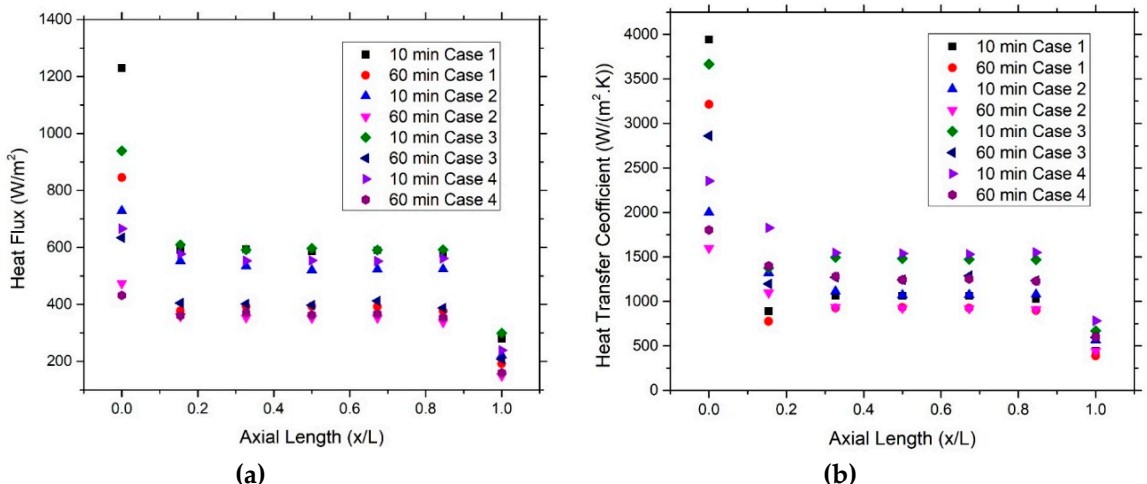

**Figure 9.** First 60-min operation meters in cases 1, 2, 3, and 4. (**a**) Heat flux; (**b**) heat transfer coefficient.

### 3.3. Effect of Plain and DDIR Coils on Ground around GHE

Figure 10a shows the temperature at a location in various elapsed times. The monitoring point (m) is around 20 mm near the pipe insulator at a depth of 1.5 m from the ground surface. The initial temperature at this point is 20.6 °C. In the first 120 min of operation, the temperature rises dramatically, at about 2.5 °C. Temperature values between variations are almost similar to each other. GHE releases much heat to the ground affect increment of the temperature in this short time range. Then, the temperature increases steadily at about 1.7 °C from 120 min until the end of operation. In turbulent flow, the temperatures of the DDIR coil and plain coil coincide with each other, and both coils have higher temperatures than others. This phenomenon confirms that in turbulent flow, both the DDIR coil and plain coil heat transfer rates have the highest values compared to other variations. In laminar flow, the DDIR coil shows a slightly higher temperature than that of the plain coil. This finding also proves that the heat transfer rate of the DDIR coil is slightly higher than that of the plain coil. In general, it can be said that an analysis of the temperature behavior in soils proves that the DDIR coil is only superior in laminar flow. The ground is unable to carry out heat recovery properly.

### 3.4. Effect of Different Materials on Plain and DDIR Coil Performance

The effect of different pipe materials, both plain-coil and DDIR-coil, is observed and compared in Figure 10b. Generally, DDIR-coil heat transfer rates tend to have a heat transfer rate greater than plain-coil in the initial period of operation. It can be observed that copper DDIR-coil has a heat transfer rate of 224 W/m compared to DDIR-coil composite at 198 W/m and DDIR-coil HDPE at 170 W/m. However, soon after that, the heat transfer rate drops rapidly. After 60 min of operation, the heat transfer rate decreases gradually due to the influence of widespread heat buildup on the soil. The use of materials that have high thermal conductivity, such as copper, is recommended as a Horizontal Ground Heat Exchanger. However, copper pipes need to be coated because copper pipes are not as flexible as HDPE. This simulation proves that higher conductivity of materials capable of obtaining higher heat transfer rates. The use of composite material that combines copper pipes and LDPE layers shows higher performance than HDPE. Although copper has a thermal conductivity of about 300 times greater than HDPE, the dominance of soil conductivity on GHE performance is enormous. The use of DDIR-coil does not show a significant difference in heat transfer rate compared to plain-coil on the same type of pipe material.

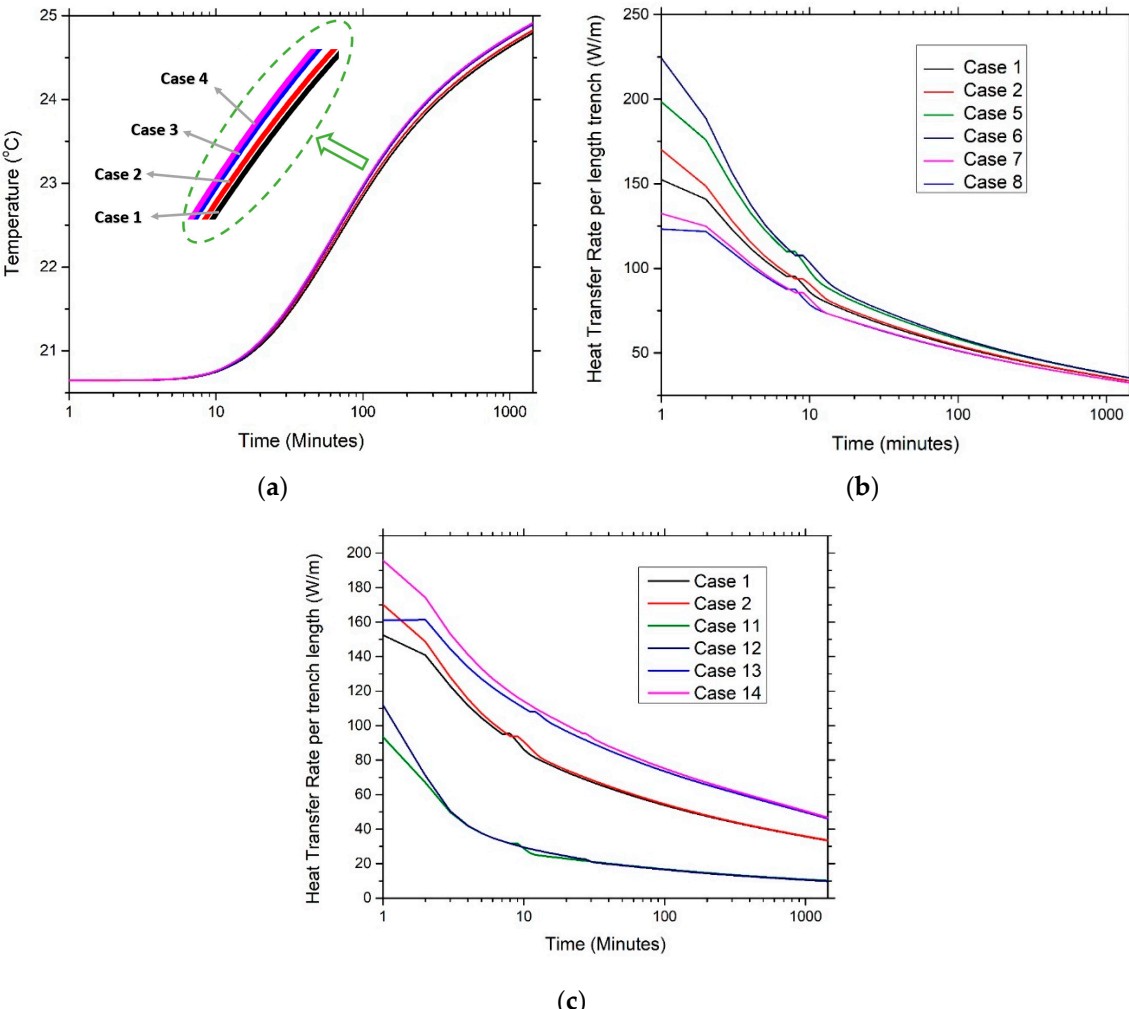

**Figure 10.** (**a**) Transient ground temperature variation in cases 1, 2, 3, and 4 at a monitoring point m at a depth of 1.5 m from the ground surface; (**b**) the effect of different pipe materials on the heat exchange rate in cases 1, 2, 5, 6, 7, and 8; (**c**) the effect of different thermal conductivities in cases 1, 2, 11, 12, 13, and 14.

## 3.5. Effect of Different Ground Thermal Conductivity

Figure 10c shows the effect of different ground thermal conductivities of sand, clay, and sandy clay on the GHE performance. Based on this figure, the heat transfer rate significantly increases with increasing ground thermal conductivity. In general, the DDIR coil has a trend of heat transfer rate that is greater than the plain coil at the beginning of operation. In the first operation time, the DDIR coil on sandy clay has a heat transfer of 196 W/m, which is the highest compared to the DDIR coil on clay of 123 W/m and the DDIR coil on sand of 112 W/m. Thermal conductivity of clay and sandy clay is approximately 300 times and 600 times that of the sand thermal conductivity, respectively. This conductivity causes the average heat transfer rate to increase by 357 times for sandy clay and 227 times for clay compared to sand on the DDIR coil. On the other hand, it can be seen that the use of the DDIR coil does not show a significant effect compared to the plain coil. This phenomenon proves that the thermal conductivity of the soil is more dominant than the convection inside the GHE.

## 4. Conclusions

This study presents the results of numerical simulations of heat extraction from two types of horizontal slinky coil, namely the DDIR coil and the plain coil in laminar and turbulent flow.

The simulation results show that the turbulent DDIR coil flow has better thermal performance than the plain coil in the first 149 min. After this time frame, the plain coil performance is superior to the end of the operating time. On the other hand, in laminar flow, the DDIR coil has a higher thermal performance than the plain coil throughout the operating period. To find out this phenomenon, we examined 60 min of operation of the two coils at several pipe locations. Based on this study, we found that in laminar flow, the average heat transfer rate for the plain coil and the DDIR coil is 59 and 60.1 W/m for 60 min of operation, respectively, whereas in turbulent flow, the average heat transfer rate is 62 and 62.3 W/m for the plain coil and DDIR coil, respectively. COP improvement factors of the plain coil and DDIR coil in laminar flow are 1.96 and 1.98, respectively. However, the COP improvement factors of the plain coil and DDIR coil in turbulent flow are 1.89 and 1.88, respectively. The difference in material types results in significant thermal performance. In the DDIR coil, copper produces heat transfer rates higher than 10 and 17 W/m, and higher than that of the composite and HDPE, respectively, in the first 60 min of operation. However, for the average heat transfer rate during 1440 min of operation, all three materials have almost the same performance. In addition, ribs and plain coils have thermal performances that coincide with each other on the same type of pipe material. Sand, sandy clay, and clay are examined to see the influence of its thermal conductivity on the GHE performance. It is found that ground conductivity is more powerful than convection in the DDIR coil regarding the heat transfer rate of the GHE.

**Author Contributions:** T.H.A. is the main author who performed the numerical modeling of the ground heat exchangers operation. A.M. provided guidance in this work, advised on the analysis of simulation results, and reviewed the manuscript. All authors have read and agreed to the published version of the manuscript.

**Funding:** This research received no external funding.

**Acknowledgments:** This research was supported by the project "Renewable energy-heat utilization technology & development project" of the New Energy and Industrial Technology Development Organization (NEDO), Japan.

**Conflicts of Interest:** The authors declare no conflict of interest.

## Nomenclature

| | |
|---|---|
| D | diameter of slinky coil (m) |
| $P_c$ | coil pitch (mm) |
| $\alpha$ | angle of ribs |
| $P_R$ | axial pitch ribs (mm) |
| H | ribs height (mm) |
| $\rho$ | density (kg/m$^3$) |
| $C_P$ | specific heat capacity (J/(kg·K)) |
| k | thermal conductivity (W/(m·K)) |
| $\lambda_{ci}$ | swirl strength (s$^{-1}$) |
| d | internal diameter tube (mm) |
| Re | Reynolds number |
| Q | heat transfer rate (W) |
| $\overline{Q}$ | heat transfer rate per length of trench (W/m) |
| Q″ | heat flux (W/m) |
| $Q_l$ | local heat transfer coefficient (W/(m$^2$·K)) |
| $\dot{m}$ | mass flowrate (kg/s) |
| x/L | nondimensional axial length of coil |
| T | temperature (°C) |
| $COP_{net}$ | Coefficient of Performance on GSHP |
| $L_{comp}$ | power input to compressor (W) |
| $\Delta T_l$ | difference between wall and bulk temperature (°C) |
| $L_{pump}$ | power input to pump (W) |

| | |
|---|---|
| $Q_C$ | cooling rate (W) |
| $COP'_{net}$ | modified $COP_{net}$ with GHE improvement |
| $Q'_C$ | increase of cooling rate (W) |
| $L'_{pump}$ | increase of pumping power (W) |
| $Q_{c\text{-}s}$ | cooling heat transfer of straight tube (W) |
| $\Delta p_s$ | pressure drop in straight tube (Pa/m) |
| $\Delta p_c$ | pressure drop in coil tube (Pa/m) |
| $\Delta p'$ | increment of pressure drop (Pa/m) |
| V | volumetric flowrate (m$^3$/s) |
| y | depth from ground surface (m) |
| Subscript | |
| i | inlet |
| o | outlet |
| b | bulk |
| w | wall |
| y | depth |

## Appendix A

Figure A1 shows the strength of the vortex on the plain coil and DDIR coil. In the plain coil, the vortex is only produced by secondary flow of the coil so that the vortex strength produced is only 3.7 s$^{-1}$. On this coil, the vortex is collected on the inner-side. The fluid moves to the inner-side coil due to centrifugal force. Then, two flow near-wall perimeters collide with each other, which leads to separation in the inner-side. Meanwhile, on the DDIR coil, the vortex is generated by a combination of secondary flow and flow generated by ribs, so that the vortex strength is 16.2 s$^{-1}$ where this value is higher than that of the plain vortex coil. Vortex location tends to be around ribs, especially in the rear ribs.

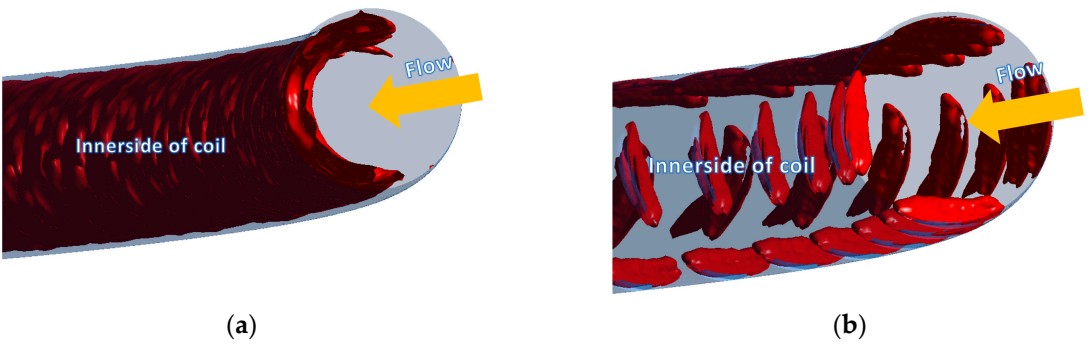

| | |
|---|---|
| (**a**) | (**b**) |

**Figure A1.** Vortex strength generated for Re = 3406. (**a**) Cross-sectional view of case 1 with $\lambda_{ci}$ = 3.7 s$^{-1}$; (**b**) cross-sectional view of case 2 with $\lambda_{ci}$ = 16.2 s$^{-1}$.

Figure A2 shows the bulk temperature and wall temperature at x/L = 0, 0.15, 0.33, 0.5, 0.67, 0.84, and 1. Similar bulk temperature and wall temperature distribution is observed in both laminar and turbulent flow in this study. In the plain coil, the wall temperature general trend showed a drastic wall temperature drop x/L = 0 to 0.15. This decrease occurs because the boundary layer of hydrodynamics has already developed while the thermal boundary layer has not yet fully developed. There is thermal interference between the GHE pipes. Then, the wall temperature slightly decrease from x/L = 0.15 to 0.33. In this section, flow changes in direction so that the flow experiences developing flow in both hydrodynamics and at the thermal boundary layer. The decrease in wall temperature occurs linearly from x/L = 0.33 to 0.84. In this section, flow has experienced both fully developed thermal- and hydrodynamics. Then, the wall temperature experienced a significant drop from x/L = 0.84 to x/L = 1. In this section, thermal interference between the GHE pipes has decreased, and the thermal boundary layer is dominated by ground heat. On the other hand, the DDIR coil shows a linear trend of wall

temperature decrease from x/L = 0 to 1. Meanwhile, bulk temperature of the DDIR coil is lower than that of the plain coil at first 10 min. However, bulk temperatures of the plain and DDIR coil coincide at 60 min.

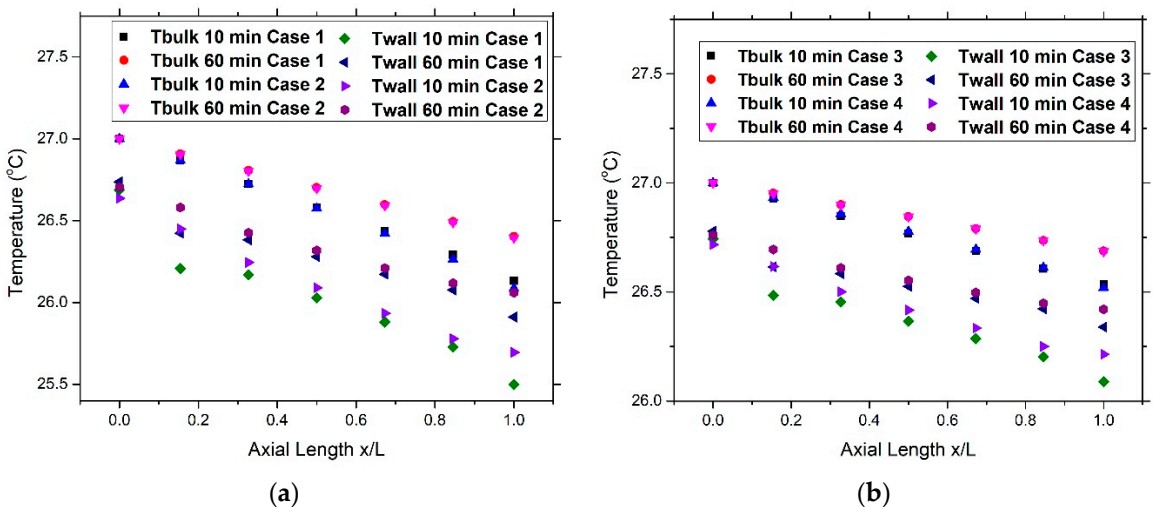

**Figure A2.** Bulk and wall temperature of first 60-min operation of the GHE. (**a**) Laminar flow, (**b**) turbulent flow.

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
