# Peer review of "Thermal Characteristics of Slinky-Coil Ground Heat Exchanger with Discrete Double Inclined Ribs"

_resources, doi:10.3390/resources9090105_

Round 1
Reviewer 1 Report
In this paper, a numerical analysis of thermal performance for
coils for application to a Ground Heat Exchanger is analyzed. In particular, the comparison between a plain coil and a DDIR-coil is discussed.
It was found that the thermal performance of DDIR-coil was
20 slightly superior to that of laminar flow plain coil.
The paper and the topic considered seems to lack of scientific value. The limitation is that though if it could be interesting to evaluate the thermal conductivity of the ground, it is not particularly relevant to the evaluation of the heat transfer coefficient inside.
The paper cannot be considered eligible for publication in a scientific review. Moreover, I have also doubts on the connection between this topic and the objectives of the review Resources. Here the authors only discuss a quite simple case, and the only real result is the calculation of a heat transfer coefficient inside the pipe. I consider this single result not sufficient for considering the paper suitable for publication.
Moreover, the form must be improved. The long section 3 is quite dispersed and the authors should be able to concentrate the results in a more efficient way, without introducing a lot of details.
I suggest the authors work on this topic increasing the elements of the discussion. For example, you can provide results considering a sensitivity analysis with respect to the variation of the ground thermal conductivity at least. Moreover, I think that you can find a better editorial collocation for such a kind of article. But the paper requires important refinements.
Reviewer 2 Report
The subject is important and current and it is relevant its investigation and The development of more efficient solutions for the Ground Heat Exchangers is crucial to make more competitive the systems based in ground source heat pumps. The work presented in the article it is interesting to be published, after minor changes:
- It would be important for the authors to state whether a mesh independence study was carried out to guarantee that the solution is independent of the mesh resolution.
- The reference in the title of Figure 5 is wrong, should be [46] instead [43];
- It is important to present a list of Nomenclature. In some equations are not identified in the text their variables (e.g. equations 1 to 7);
- The ribs can increase thermal performance but they do introduce pressure drop. It is always important to assess both effects. Through CFD simulation it is easy to obtain the respective values. It would be important to present the difference in pressure drop between ribs coil and plain coil, or if they are negligible, mention this.
Reviewer 3 Report
Here are the most important topics/questions to be dealt with:
1/ Novelty of work should be highlighted in proper way.
2/ How were estimated the properties of clay? They are very important and useful information for readers. Thus, subsection 2.1 should rewritten by the Authors.
3/ All schemes of mesh have to depict in this work. Otherwise, without this information, the article / the numerical model is devoid of cognitive value.
4/ Mesh model and grid independence checking should be detailed presented in this work.
5/ Due to the fact that the work concerns simulation, Reviewer did not find the reply to question: What error formula was used to estimate rightness of calculations?
6/ Notation of unit for temperature needs improvements. 27 C is not the same like 27oC. The same problem goes for the average heat transfer coefficient (see Page 11, line 288) and also Page 18 lines 422, 424). W/m2-K is not the same like W/(m2K).
7/ All symbols used in the equations should be explained directly where cited, for instance equation 8.
8/ What does mean solid black line in figure 4?
9/ Symbol "lambda" for local heat transfer coefficient is vague. In literature “lambda” usually refers to thermal conductivity coefficient or wavelength of light beam.
10/ Why did you depict in the draft manuscript only selected simulation results?
11/ Figure 8 is illegible and needs improvements. All curves should be better depicted.
12/ Typing suggestions:
Page 1, line 40: GHSP => GSHP
Page 5, line 126: boundary condition => boundary conditions
initial condition => initial conditions
Page 7, line 166: initial condition => initial conditions
Page 10, line 241: Flow => flow
On the basis of above considerations, minor revision of draft manuscript is suggested.
Reviewer 4 Report
The authors proposed the CFD simulation on the slink-coiled ground heat exchangers. The manuscript was well-organized; adding the ribs on the coil surface was interesting; the results and discussion section was good; however, the reviewer suggests a modification for the possible publication in the Resources as below:
Based on the proposed results, it seems that adding the ribs on the coil surface could enhance the heat transfer, but it was insignificant. Of course, the slight improvement of the heat transfer also contributes to the scientific society, but the ribs also increase the frictional pressure drop leading to the energy loss, especially as the coil length increases more. Even though the others [31-34] proposed that the energy loss by the pressure drop was insignificant, the reviewer thinks that the submitted manuscript should include the pressure drop results, so that the design of the proposed configurations was enough to use in practical.
Round 2
Reviewer 1 Report
Some of my comments have been taken into account but some have b
The authors have worked on the paper. Concerning my comments, some of them have been taken into account but some have been treated very briefly. I think that some further refinements are always necessary before the publication.
In principle, I am not really in agreement with the authors about the response: "We think that the research topic and objective review resources journal is connected. The authors elaborate on the application of Ground Heat Exchanger to explore geothermal energy resources". I think that the ground heat exchangers are not connected with geothermal resources but is only a method for "energy-saving". So this is an indirect use of geothermal energy.
Concerning the observation on the actual version of the paper, I think that section 3 is confused. In the actual version, there are too many subsections (up to 8) and it is full of unnecessary details. Please try to reduce this section of factor 2 and reduce the number of figures. In the end, I suggest that the maximum number of figures could be 10 maximum.
My last observation is about section 2. Now the first 6 equations seem to be extrapolated from Ansys Fluent Manual and are not necessary. You can simply enumerate the equations without the formulas.
Finally, I suggest carefully read the final version of the paper. There are a lot of typos and parts that require corrections. For example, Conclusions are considered section 3, while I think that they are section 4.
With those further careful modification, I think that the paper could be eligible for publication and I will express this opinion.
Round 3
Reviewer 1 Report
I have noticed that in the third version the authors have worked on a great part of my concerns. But the paper is not yet suitable for publication because before some additional corrections and integration are required. The authors must adjust some of the problems before the final publication.
My selection is of "major revision" but this is only because I think that it could be important a further revision by the authors, but after this additional work the paper will be suitable for publication.
I enumerate the more relevant observations
1) The authors have correctly removed the old equations (1-6) that are available in the manual of the commercial software. But now the number of equations start from (7) and I think that can adjust it both in the sequence and in the text
2) The second observation is relative to section 3. I have suggested to them to better rearrange it because in the old version it was very confused and with too much figures. The authors have tried to remove some figure (not so much) but the organization of the section is not optimal. I think that before introducing the various sections 3.1, 3.2 and so on, a short (10 lines or something like that) introduction to the results
3) The authors give emphasis to the data with too much details.. This is not good when the error on the analysis are comparable with the accuracy level, it could be not particularly relevant to introduce some values. For example what is the practical relevance of introducing the value of 60.02 W/m. Who is able to appreciate the value of 0.02 W/m. This value has only a numerical sense
4) In the new version, for the relevant modification introduced, there are several typographical mistakes. I suggest to check carefully a "clean version of the paper" after introducing the new modifications
